biomaterials

DOPA, SCPP, bone cement, bone repair

**Author for correspondence:**
Xixun Yu
e-mail: yuxixun@163.com

†These authors contributed equally to the study. This article has been edited by the Royal Society of Chemistry, including the commissioning, peer review process and editorial aspects up to the point of acceptance.

# A promising material for bone repair: PMMA bone cement modified by dopamine-coated strontium-doped calcium polyphosphate particles

Xing Liu[1,†], Can Cheng[1,†], Xu Peng[1,2], Hong Xiao[3], Chengrui Guo[1], Xu Wang[1], Li Li[4] and Xixun Yu[1]

[1]College of Polymer Science and Engineering, Sichuan University, No. 24 South Section 1, Yihuan Road, Chengdu 610065, People's Republic of China
[2]Laboratory Animal Center, Sichuan University, Chuanda Road, Chengdu 610065, People's Republic of China
[3]Department of Pain Management, West China Hospital, Sichuan University, No. 37, GuoXue Xiang, Chengdu 610041, People's Republic of China
[4]Department of Oncology, The 452 Hospital of Chinese PLA, No. 317, Shunjiang Road, Chengdu, Sichuan Province 610021, People's Republic of China

XY, 0000-0002-2301-8727

Polymethyl methacrylate (PMMA) bone cement has been widely used in clinics as bone repair materials for its excellent mechanical properties and good injection properties. However, it also has defects such as poor biological performance, high temperature, and the monomer has certain toxicity. Our study tried to modify the PMMA bone cement by doping with various particle weight fractions (5, 10 and 15%) of SCPP particles and polydopamine-coated SCPP particles (D/SCPP) to overcome its clinical application disadvantages. Our study showed that all results of physical properties of samples are in accordance with ISO 5833. The 15% D/SCPP/PMMA composite bone cement had much better biocompatibility compared with pure PMMA bone cement and SCPP/PMMA composite bone cement due to the best cell growth-promoting mineralization deposition on the surface of 15% D/SCPP/PMMA composite bone cements and $Sr^{2+}$ released from SCPP particles. Our research also revealed that the reaction temperature was found to be reduced with an increase in doped particles after incorporating the particles into composite bone cements. The novel PMMA bone cements modified by D/SCPP particles are promising materials for bone repair.

# 1. Introduction

Polymethyl methacrylate (PMMA) bone cement has been used in clinics for its excellent mechanical properties and simple operation procedure. However, there are still defects such as poor biocompatibility and overheating resulting from solidification. Moreover, PMMA bone cement cannot be degraded, lacks biological activity, and cannot form osseous bonding with host bone tissue. After being implanted *in vivo*, it is poor in integrating with the surrounding bone tissue and not conducive to bone cell adhesion and growth.

Calcium polyphosphate (CPP) is a new type of bioceramic for bone repair and has attracted extensive attention in bone repair; both *in vitro* and *in vivo* tests have confirmed that CPP can promote bone growth [1,2]. Strontium (Sr), the homologous element of calcium, presents a dual action of improving bone formation as well as inhibiting bone resorption. Strontium-doped calcium polyphosphate (SCPP) was prepared by introducing strontium into CPP. Our previous studies showed that SCPP can significantly promote the growth of osteoblasts, and it also had a good stimulatory effect on the secretion of angiogenic growth factors (including vascular endothelial growth factor (VEGF) and basic fibroblast growth factor (bFGF)) by osteoblasts [3,4]. The neovascularization induced by angiogenic growth factors can increase the blood supply of bone interface [5], which is beneficial to the growth of the interfacial bone. Among them, the SCPP containing 8% Sr has the best effect.

Many studies have shown that the addition of bioactive ceramic particles (as fillers) to PMMA bone cement can improve the biological performance of PMMA bone cement and reduce the reaction temperature [6–8]. It is a good way to modify the biological properties of PMMA bone cement, but there is still a disadvantage that the interfacial compatibility between bioactive ceramic particles and PMMA bone cement is bad in this modification. So, a stable modification structure through the ceramic particles/PMMA interface is difficult to be formed, and the relevant performance characteristics of this modified PMMA bone cement are reduced. This limits its further application. Therefore, finding an effective binder to increase the connection between ceramic particles and PMMA is an effective way to improve the interfacial compatibility between these two materials.

To address the disadvantage mentioned above, miraculous biomolecule DOPA [9] was introduced to modify the surface of ceramic particles to improve the interfacial compatibility with PMMA [10]. In recent years, DOPA has been recognized as a useful and promising adhesion molecule for its easy operation, solvent-free, non-toxic and excellent adherent properties, and has attracted increasing attention [11]. DOPA is easy to self-polymerize, forming polydopamine, which possesses a large amount of active functional groups and can react with most materials [12]. It can be inferred that DOPA modification would increase the connection between ceramic particles and PMMA. Although DOPA has been used as an ideal adhesion molecule to modify the surface of many biomaterials, there were still no reports about using DOPA as the adhesion molecule to modify the surface of ceramic particles to enhance ceramic particles–PMMA interface compatibility of the ceramic particles/PMMA composite bone cements in present studies.

In this study, the 8% SCPP particles were employed as the bioactive ceramic fillers to modify PMMA bone cements. DOPA was first introduced to coat the surface of 8% SCPP particles as an intermediary agent, and then these DOPA modified-SCPP particles were incorporated into PMMA bone cements. After that, the biological and physico-chemical properties of this 8% SCPP particles/PMMA composite bone cement were systematically researched (figure 1).

# 2. Materials and methods

## 2.1. Preparation of SCPP/PMMA and D/SCPP/PMMA composite bone cements

After 8% β-SCPP particles were coated by DOPA, the SCPP particles and D/SCPP particles were admixed to pre-polymerized beads PMMA using a mixer to prepare the D/SCPP particles–PMMA beads combinations or SCPP particles–PMMA beads combinations. Subsequently, the combinations mentioned above were admixed to the monomer liquid and stirred slowly to reduce the amount of the air bubble inclusion to obtain SCPP/PMMA and D/SCPP/PMMA composite bone cements.

## 2.2. Characterizations of SCPP/PMMA and D/SCPP/PMMA composite bone cements

EDS test, pore size distribution and porosity of PMMA composite bone cements, and dispersion of SCPP and D/SCPP particles in PMMA matrix mainly confirm whether and how the SCPP and D/SCPP particles are successfully doped into the PMMA matrix. Since the weight fraction of particles doping in composite bone

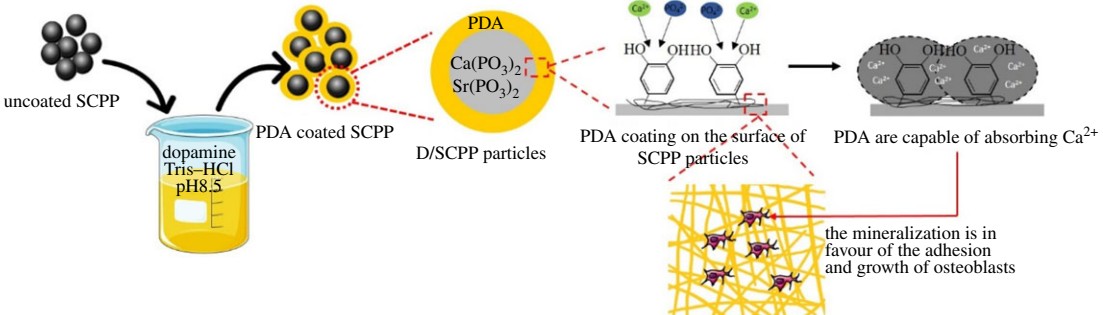

**Figure 1.** Preparation, characterization, physico-chemical properties and bioactivity of composite bone cements.

cements has no effect on its characterizations, 15% D/SCPP/PMMA composite bone cement material is selected as representatives for these tests to show whether Sr element and DOPA are doped into PMMA.

## 2.3. Determination of physical properties of SCPP/PMMA and D/SCPP/PMMA composite bone cements

The physical properties of SCPP/PMMA and D/SCPP/PMMA composite bone cements test, including dough time, intrusion, setting time, maximum temperature, biomechanical properties and *in vitro* mineralization performance were carried out according to ISO 5833. See the electronic supplementary material for detailed test methods.

## 2.4. Effect of D/SCPP/PMMA on the biological activity of osteoblast cells

In the following research, we prepared pure PMMA bone cement and various composite bone cements including SCPP/PMMA and D/SCPP/PMMA with SCPP particles weight fractions of 5, 10 and 15%. The effects of these samples on the biological activity of osteoblast cells were investigated.

Fourth passage osteoblastic cells (MG63) were purchased from West China Hospital, Sichuan University (China). MG63 were cultured on the bone cement samples ($\Phi10 \times 2$ mm) put in a 24-well culture plate at an initial density of $5 \times 10^3$ cells per specimen. The cells were cultured at 37°C in a humidified environment. The proliferation of MG63 on all bone cement specimens was estimated using MTT assay at 1, 4, and 7 days, respectively. The cell adhesion and morphology were observed by SEM and immunofluorescence staining at 4 days.

To further research the effect of D/SCPP/PMMA and SCPP/PMMA composite bone cements on osteogenesis process, the human bone morphogenetic protein 2 (BMP-2), VEGF and alkaline phosphatase (ALP) which secreted from co-cultured MG63 and cumulative in culture medium were detected according to the manufacturer's instructions (R&D Corp.). Based on it, the optimal doping amount of SCPP particles was determined.

## 2.5. The detection of $Sr^{2+}$ released from cell–cement constructs in cell culture mediums

To explore the promotion of $Sr^{2+}$ released from SCPP/PMMA and D/SCPP/PMMA to MG63 cell proliferation, $Sr^{2+}$ in culture medium was determined using ICP detection. The culture mediums were diluted by distilled water and the concentrations of $Sr^{2+}$ were tested by ICP spectrometer.

## 2.6. Statistical analysis

SPSS (v. 19.0) was used for statistical analysis. Quantitative data were expressed as a mean value with its standard deviation indicated (mean ± s.d.). To determine the differences in measured properties of various groups, it was analysed by one-way ANOVA for statistical significance, and the statistical significance was signed at $p < 0.05$.

# 3. Results and discussion

## 3.1. The pore size distribution and porosity of PMMA composite bone cements

SEM showed that after the incorporation of SCPP and D/SCPP particles into the PMMA matrix, the materials had some partial defect structures (pore and concave–convex structures). These structures resulting from the

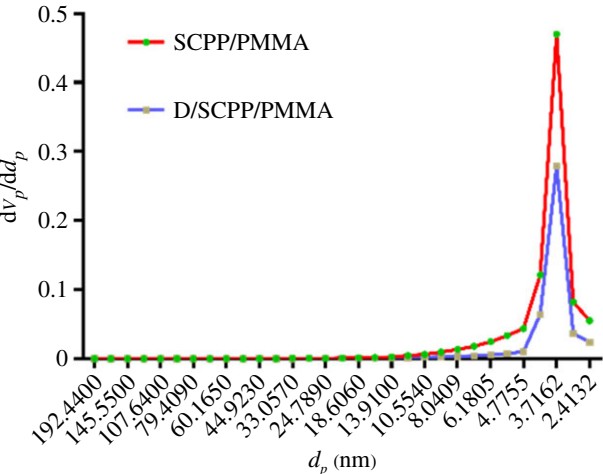

**Figure 2.** Pore size distribution of SCPP/PMMA and D/SCPP/PMMA.

exfoliation of agglomerated particles could be observed at the fracture surface of SCPP/PMMA and D/SCPP/PMMA (electronic supplementary material, figure S3). However, both agglomeration of SCPP particles and formation of depression structures at the fracture surface of D/SCPP/PMMA significantly decreased compared with SCPP/PMMA. These results indicated that DOPA increased the connection between SCPP particles and PMMA, and improved the interfacial compatibility between these two materials. We could infer that the D/SCPP/PMMA composite bone cements had better mechanical properties than the SCPP/PMMA composite bone cements. The pore size distribution of SCPP/PMMA and D/SCPP/PMMA composite bone cement were also measured, and the pore size ranged from 3 to 4 nm (figure 2), the average diameters of these pores were 3.5243 and 3.2511 nm (electronic supplementary material, table S2), respectively. Therefore, the obtained SCPP/PMMA and D/SCPP/PMMA composite bone cements were mesoporous materials. Some studies have shown that mesoporous bioglass materials could form a good layer of apatite mineralization in SBF, thus promoting osteoblast adhesion and proliferation [13,14]. The nanometre-size pores on the material surface could increase the surface area, thereby increasing the binding sites of the protein and facilitating the adhesion of cells through the protein layer. Moreover, numerous studies suggested that the morphological and chemical characteristics of nanostructures were close to natural bone tissue, which not only helped the osteoblast pseudopodia to migrate through the pores of the materials but also contributed to the proliferation of new bone tissue, played a role in bone conduction and provided a more ideal growth environment for bone tissue regeneration [15,16].

## 3.2. Dispersion of SCPP and D/SCPP particles in PMMA matrix

The micro-CT image (figure 3) showed that the yellow parts were doped SCPP and D/SCPP particles. As shown in the figure, the particles were dispersed evenly in the PMMA matrix, but the particle size was not uniform and agglomeration occurred to some extent. However, the particle agglomeration in D/SCPP/PMMA was reduced compared with SCPP/PMMA. We could speculate that DOPA coating made the connection between inorganic particles and PMMA matrix tighter; the interfacial compatibility was thus improved, which made it easier for particles to disperse into the matrix. Therefore, the particles presented less agglomeration in D/SCPP/PMMA, and D/SCPP/PMMA composite bone cements presented better mechanical property.

## 3.3. Determination of physical properties of SCPP/PMMA and D/SCPP/PMMA composite bone cements

### 3.3.1. Dough time, intrusion, setting time and maximum temperature

The dough time of SCPP/PMMA and D/SCPP/PMMA composite bone cements was shorter than PMMA bone cements, and they decreased as the content of doped particles increased (figure 4a). The dough time of all PMMA composite bone cements was less than 5 min. The intrusion test results of SCPP and D/SCPP particles was lower than PMMA bone cements, and with the increase of doping amount, the penetration of bone cements decreased gradually (figure 4b). Figure 4c shows the setting

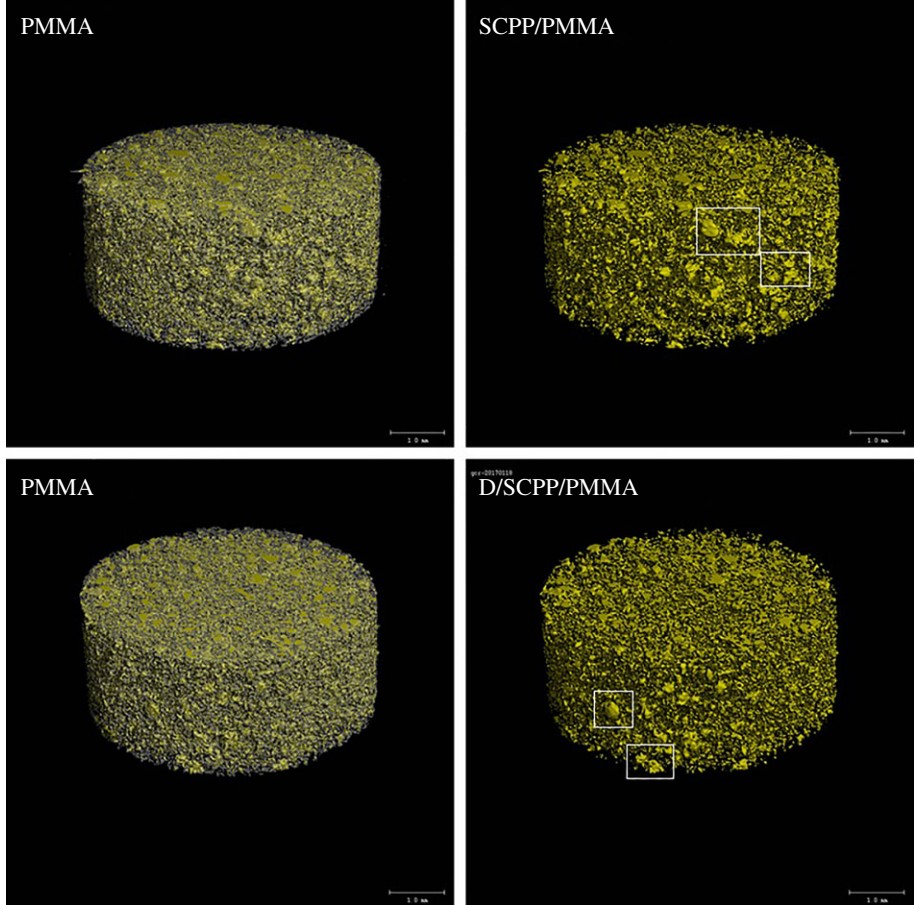

**Figure 3.** Micro-CT of PMMA bone cement composite (white boxes indicate agglomeration).

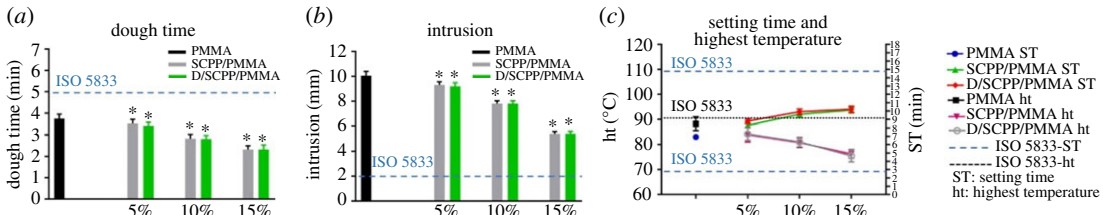

**Figure 4.** The dough time, intrusion, setting time and highest temperature of PMMA bone cement: (*a*) dough time; (*b*) intrusion; (*c*) setting time and highest temperature (*significant difference with PMMA group; $p < 0.05$).

time and the highest reaction temperature of PMMA composite bone cements. After incorporating the particles into composite bone cements, the setting time was found to be increased with an increase in doped particles. Meanwhile, the reaction temperature was also reduced. The higher the amount of particles were doped in composite bone cements, the more the reaction temperature reduced. The 15% SCPP/PMMA and 15% D/SCPP/PMMA groups had the longest setting time (9.7 min) and the lowest maximum temperature (75.5°C). All results met and were within the ISO 5833 standard range, indicating that D/SCPP/PMMA composite bone cements have a good applicability.

The curing reaction process of PMMA bone cement is a process of a spontaneous chemical reaction when the initiators in the solid phase are admixed to the liquid phase monomer [17]. In this process, the polymerization reaction proceeds continuously, the viscosity of the system increases, the reaction exotherm and the temperature of the system also continuously increase. When the particles are incorporated, the power/liquid ratio of the entire system increases, and the viscosity of the system also increases; therefore, the dough time and intrusion of the composite bone cements decrease as the amount of particles doping in composite bone cements increases.

With the increase of doping amount of SCPP and D/SCPP particles, the amount of PMMA matrix in the bone cement system decreased, the presence of particles affected the polymerization of PMMA, and

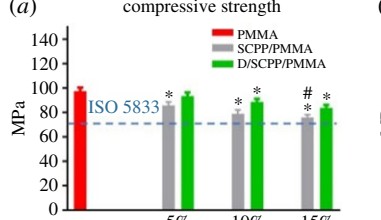
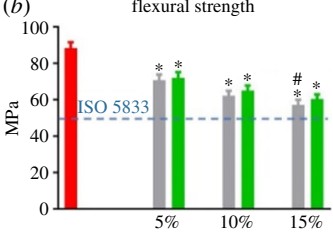
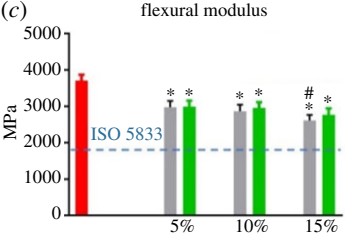

**Figure 5.** Mechanical properties of PMMA bone cement: (a) compressive strength, (b) flexural strength, (c) flexural modulus (*significant difference with PMMA group; # significant difference with 15% D/SCPP/PMMA group; $p < 0.05$).

the absorption of heat by particles caused the reduction in reaction temperature of the bone cement system. These might make the reaction slow, resulting in a longer setting time of the composite bone cement and a decrease in the highest reaction temperature. In clinical applications, the setting time represents the time when bone cement is heated up. If the setting time is too short, the operation time will not be enough [18]. As a result, the bone cement has not yet reached the ideal position and its viscosity has increased, seriously affecting the application of the bone cement in surgery. However, if the setting time is too long, it will cause the solidification of bone cement to be slow, resulting in the flowing out of slurry after implantation and affecting the surrounding tissue [19]. Excessively high reaction temperature of bone cement will burn the surrounding tissue in implant site [20], affect tissue-compatibility of bone cement, and reduce the success rate in related surgery [21].

### 3.3.2. Mechanical properties test

The mechanical properties of PMMA bone cement, including compressive strength (figure 5a), flexural strength (figure 5b) and flexural modulus (figure 5c) [22], were significantly higher than SCPP/PMMA and D/SCPP/PMMA composite bone cement and decreased with an increase in the doping amount of particles, but all mechanical test results met ISO 5833 standards.

Studies have shown that when bioactive ceramic particles are incorporated into PMMA bone cement matrix, the particles will inevitably agglomerate. The agglomerated regions appear cracked and gradually expand, which leads to the impairment and reduction of material's mechanical properties, when the entire materials were subjected to external forces [13,23]. Compared with SCPP/PMMA, the mechanical properties of D/SCPP/PMMA were better, because the coating of DOPA could increase the interfacial compatibility of the two phases [24], increase the binding force between SCPP particles and PMMA matrix, and then improve the mechanical properties of bone cement to some extent.

### 3.3.3. *In vitro* mineralization performance of bone cement

As shown in electronic supplementary material, figure S4 of SEM image, after immersing in SBF for 6 days, the thin precipitated layers were observed on the surface of PMMA, SCPP/PMMA and D/SCPP/PMMA composite bone cement. According to the results of elements analysis for these precipitated layers (electronic supplementary material, figure S5), we could infer that the precipitated layers were mainly Ca–P deposition. The contents of Ca and P in precipitated layer formed on the surface of D/SCPP/PMMA composite bone cement were higher than that of PMMA and SCPP/PMMA groups (electronic supplementary material, table S3), which indicated the best mineralization deposition on the surface of D/SCPP/PMMA composite bone cement *in vitro*. The reasons for this are as follows.

The mineralization deposition on the surface of SCPP/PMMA composite bone cement was better than that of the PMMA group due to the high contents of Ca and P on the surface released from SCPP incorporated into PMMA. On this basis, after being coated with DOPA, abundant phenolic hydroxyl group and thiol group in DOPA terminal region coated on the surface of SCPP particles were capable of absorbing $Ca^{2+}$ in SBF and then formed a complex compound which further adsorbed $PO_4^{3-}$ in solution. All these actions promoted the best mineralization deposition on the surface of D/SCPP/PMMA composite bone cement [25]. The mineralization depositions on the surface of composite bone cements are in favour of the adhesion and growth of osteoblast cells on it and then promote osteogenesis.

### 3.4. Effect of D/SCPP/PMMA on the biological activity of osteoblast cells

In the following research, we prepared pure PMMA bone cement and various composite bone cements including SCPP/PMMA and D/SCPP/PMMA with SCPP particles weight fractions of 5, 10 and 15%.

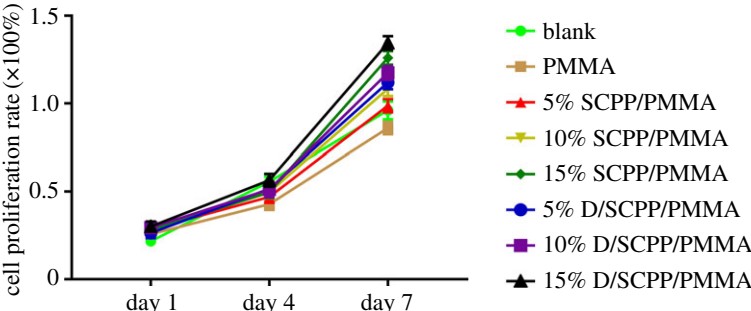

**Figure 6.** The MG63 cell proliferation rate after culturing with PMMA bone cement for 1, 4 and 7 days.

The effects of these samples on biological activity of osteoblast cells were investigated. Based on it, the optimal doping amount of SCPP particles was determined.

### 3.4.1. The proliferation of osteoblast cells cultured on bone cements

The cell proliferation rates of MG63 cultured on various bone cement specimens for 1, 4 and 7 days are shown in figure 6. Compared with pure PMMA bone cement, all SCPP/PMMA and D/SCPP/PMMA composite bone cements could promote the growth of MG63. During the culture duration, in promoting MG63 proliferation, the role of specimens in SCPP/PMMA and D/SCPP/PMMA groups was more and more obvious with the increase in the doses of doped SCPP particles. MG63 cultured on D/SCPP/PMMA samples showed a better proliferation rate than those in SCPP/PMMA groups on the seventh day. This might be because the cell growth-promoting mineralization deposition on the surface of D/SCPP/PMMA composite bone cements was best. Furthermore, 15% doping amount of SCPP particles showed the optimal effect, which suggested it was the best dose.

### 3.4.2. Adhesion and proliferation of osteoblast cells on the surface of cement

SEM micrographs of adhesion and growth of MG63 cultured on different composite bone cements on the fourth day are shown in electronic supplementary material, figure S6. As exhibited, MG63 could adhere and grow well on the surface of all bone cements (including PMMA, 15% SCPP/PMMA and 15% D/SCPP/PMMA). A lot of spreading MG63 was observed on the surface of all bone cements, especially the MG63 on the surface of 15% D/SCPP/PMMA composite bone cement appeared to grow better, and they connected each other tightly via extended pseudopodia. Some MG63 were observed to be well attached to mineralization deposition on the surface of 15% D/SCPP/PMMA composite bone cement and grew in three dimensions. The growth of MG63 cultured on the surface of various bone cements was also examined by fluorescent staining with Phalloidin and DAPI (as shown in figure 7). The result showed that the quantity of nuclei distributed on the surface of 15% D/SCPP/PMMA composite bone cements exceeded that on the surface of samples in the other two groups, which means more living MG63 was attached to 15% D/SCPP/PMMA composite bone cements. Furthermore, after the cytoplasm was further stained by rhodamine phalloidin, the cell morphology and filamentous actin (f-actin) were clearly presented. All these results indicated that the mineralization deposition (calcium/phosphorus deposits) had good effects on the adhesion and proliferation of osteoblast cells.

### 3.4.3. Detection of the ALP, BMP-2 and VEGF expression of osteoblastic cells co-cultured with bone cement specimens

The secretion of the ALP, BMP-2 and VEGF from co-cultured MG63 and cumulative in culture medium is shown in figure 8. With the increase of doping amount SCPP particles, the secretion of ALP, BMP-2 and VEGF from MG63 co-cultured with bone cement specimens increased correspondingly. Among them, MG63 cultured on 15%D/SCPP/PMMA composite bone cements presented the maximal expression level of three cytokines. The reason for this might be as follows. As mentioned in §3.4.1, MG63 cultured on 15% D/SCPP/PMMA composite bone cement showed a better proliferation rate than those in other groups, then there were more cells to secrete the cytokines on 15% D/SCPP/PMMA samples.

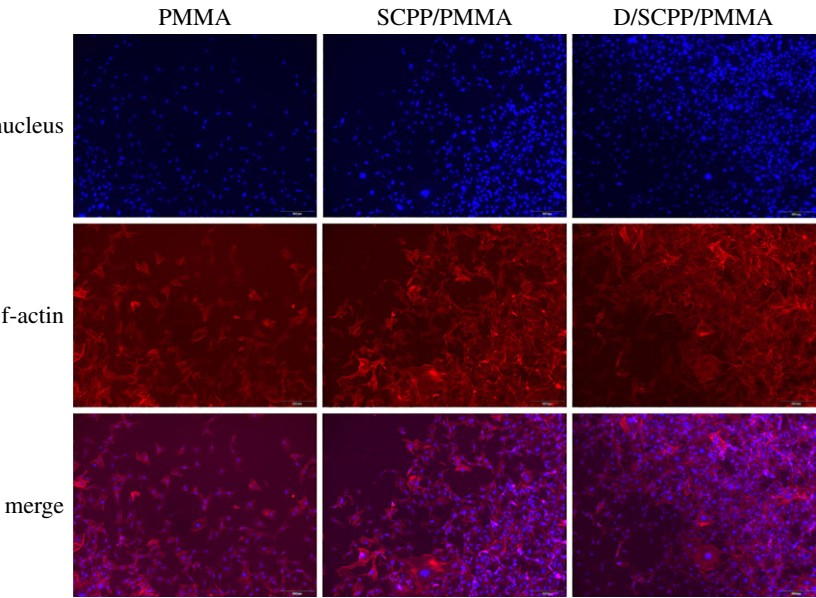

**Figure 7.** The confocal microscopy images of adhesion and growth of MG63 cells on PMMA composite bone cement.

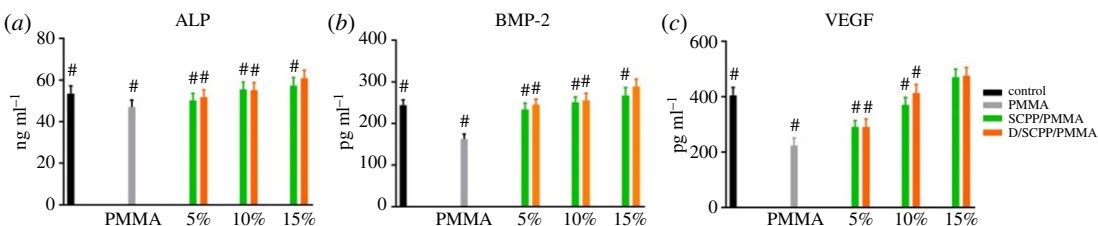

**Figure 8.** The effect of various bone cement specimens on (*a*) ALP, (*b*) BMP-2 and (*c*) VEGF secretion from MG63 cultured with materials ([#]significant difference with 15% D/SCPP/PMMA group; $p < 0.05$).

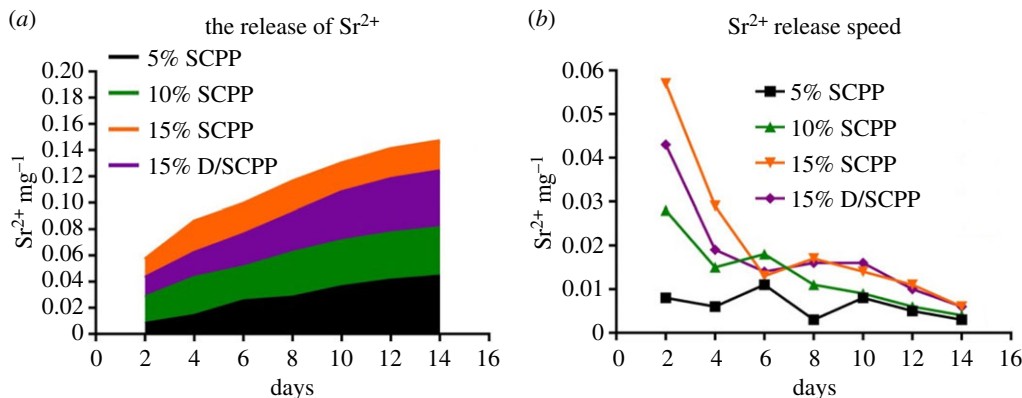

**Figure 9.** Cumulative amount of $Sr^{2+}$ ions released from various samples during 14 days (*a*) and the $Sr^{2+}$ release rate of various samples co-cultured with MG63 (*b*).

ALP provides the necessary phosphoric acid for the deposition of hydroxyapatite in the process of bone formation, which is beneficial to the process of osteogenesis. It is expressed during the post-proliferative period of extracellular matrix maturation and has been widely recognized as a marker for osteoblast differentiation. BMP-2 is the most important bone formation regulator and has a very good effect in inducing osteogenesis [26,27]. VEGF is a very important angiogenic growth factor that activates the vascular endothelial cells in the tissues surrounding and promotes the formation of tubular structures [28,29], which is important in the entire bone formation process. It is concluded that 15% D/SCPP/PMMA composite bone cement shows a notable ability to stimulate osteogenesis,

which plays a crucial role in the formation of new bone. In addition, the active factors BMP-2 and VEGF secreted by cells partially adhere to the surface of bone cements, which can improve the adhesion and growth of MG63 on the surface of the materials.

## 3.5. The release of $Sr^{2+}$

Within 14 days, $Sr^{2+}$ was continuously released from composite bone cements with increasing culture time (figure 9a), but the release rates appeared to be gradually steady and slow overall (figure 9b). Meanwhile, the higher the amount SCPP particles were doped into PMMA, the higher the amount of $Sr^{2+}$ was released from composite bone cement, and the amount of $Sr^{2+}$ released from various samples was 15% SCPP > 15% D/SCPP > 10% SCPP > 5% SCPP. Due to the significant promotion of mineralization deposition (calcium/phosphorus deposits) by the DOPA coated on the surface of SCPP particles, the release amount of $Sr^{2+}$ from 15% D/SCPP/PMMA was less than 15% SCPP/PMMA. The Sr element is a normal component of human bones and teeth [30,31]. Some research has found that Sr could promote MSCs to differentiate into osteoblast cells, enhance bone mass, inhibit osteoclast cells activity while inhibiting bone resorption [32]. The release profile of $Sr^{2+}$ from various samples in our research was basically consistent with the proliferation curve of MG63 cells, MG63 cultured on all SCPP/PMMA and D/SCPP/PMMA composite bone cements showed better proliferation rate than those on pure PMMA bone cements, and the cell proliferation-promoting effect of bone cement became more and more obvious with the increase in the doses of doped SCPP particles. Therefore, it has been proved again that $Sr^{2+}$ can promote the proliferation of osteoblasts to a certain extent.

## 4. Conclusion

To address some disadvantages of PMMA bone cement, such as poor biocompatibility and overheating resulting from solidification, in this study, the 8% SCPP particles and D/SCPP (8% SCPP particles coated by DOPA for improving interfacial compatibility between bioactive ceramic particles and PMMA bone cement) were respectively employed as the bioactive ceramic fillers to modify PMMA bone cements at particle weight fractions of 0, 5, 10 and 15%. Our data demonstrated that 15% D/SCPP/PMMA composite bone cement had much better biocompatibility compared with pure PMMA bone cement and SCPP/PMMA composite bone cement. It could not only promote the adhesion and proliferation of osteoblast cells, but stimulate the secretion of osteogenic growth factors which are confirmed to promote bone formation. Our research also revealed that the reaction temperature was found to be reduced with an increase in doped particles after incorporating the particles into composite bone cements. Therefore, we propose that PMMA bone cements modified by D/SCPP particles represent a potential candidate material for bone repair.

Ethics. This study was approved by the Medical Ethics Committee of Sichuan University (ref. no. 2019074A). All participants gave their written informed consent prior to testing.
Data accessibility. The datasets supporting this article have been uploaded as part of the electronic supplementary material, including Experimental method and Results.
Authors' contributions. X.L. and C.G. performed the experiments and drafted the manuscript with the support from C.C. and X.W.; X.P. participated in data analysis; H.X. and L.L. participated in the design of the study; X.Y. is responsible for research design and results analysis. X.L. and C.C. contributed equally to this manuscript. All authors gave final approval for publication.
Competing interests. We have no competing interests.
Funding. This work was financially supported by the National Key R&D Program of China (grant nos. 2016YFC1100900, 2016YFC1100901, 2016YFC1100903, 2016YFC1100904), the Key Research and Development Program of Sichuan Province (grant no. 2019YFS0121), Chengdu Science and Technology Huimin Project (grant no. 2016-HM01-00273-SF).
Acknowledgements. We are grateful to anonymous reviewers.

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
