## [Reviewer comments · Royal Society Open Science]

Review History

RSOS-191028.R0 (Original submission)

Review form: Reviewer 1 (Gamze Kuku)

Is the manuscript scientifically sound in its present form?

Yes

Are the interpretations and conclusions justified by the results?

Yes

Is the language acceptable?

Yes

Do you have any ethical concerns with this paper?

No

Have you any concerns about statistical analyses in this paper?

No

Recommendation?

Accept with minor revision (please list in comments)

Comments to the Author(s)

The manuscript describes coating of DOPA as an additional layer on SCCP/PMMA bone cements to improve biocompatibility and stability issues of bone cements that can be used in bone repair. The production method and characterization data is satisfactory and the manuscript can be published in RSOS. However, there minor points to consider about the figures.

Figure 4, 5, 8 is too small to be properly seen. Statistically significant data sets are difficult to be observed from the figures, too.

Figure 7 is it a SEM or confocal microscopy image? Scale bars are impossible to be seen.

Review form: Reviewer 2

Is the manuscript scientifically sound in its present form?

No

Are the interpretations and conclusions justified by the results?

No

Is the language acceptable?

Yes

Do you have any ethical concerns with this paper?

No

Have you any concerns about statistical analyses in this paper?

No

Recommendation?

Reject

Comments to the Author(s)

The novelty of this manuscript is sort of low. Authors lacked of tests in characterizing the synthesized materials and showing enough biological effects, leaving a superficial analysis far from completion. Animal models should be examined and included in this study as in vitro cell study could not convince the suitability of this materials for bone repair. For this reason, I do not see many novelty in this manuscript.

Review form: Reviewer 3 (Wenbo Zhou)

Is the manuscript scientifically sound in its present form?

Yes

Are the interpretations and conclusions justified by the results?

Yes

Is the language acceptable?

Yes

Do you have any ethical concerns with this paper?

No

Have you any concerns about statistical analyses in this paper?

No

Recommendation?

Accept with minor revision (please list in comments)

Comments to the Author(s)

Review of "A promising material for bone repair: PMMA Bone Cement Modified by Dopamine-coated Strontium-doped Calcium Polyphosphate Particles" for Royal Society Open Science:

This manuscript reported a new type of biofunctional composite composing DSCPP PAs and PMMA matrix, and evaluated its physicochemical properties and bio-activity. It requires some revisions before acceptance. Below are some of my comments.

1. In the summary, "...the reaction temperature was found to reduce with an increase..." should be "...the reaction temperature was found to be reduced with an increase...". Please also check the grammar in the context.
2. In the introduction, the authors stated that 8% SCPP was used, while in the material section, 5%, 10%, 15% were tested. These information is inconsistent.
3. Image resolution needs improving. For example, it is difficult to read the labels in Fig. 4.
4. The authors inferred that the D/SCPP/PMMA composite bone cements had better mechanical properties than the SCPP/PMMA composite bone cements. However, in Fig 5, their flexural strength/modulus is similar.
5. In the results, 15% doping was demonstrated to be the optimal for the biological activity. However, in the summary section, the authors stated that "the biological and physicochemical properties of this 8% SCPP particles/PMMA composite bone cement were systematically researched".

Decision letter (RSOS-191028.R0)

22-Jul-2019

Dear Professor Yu:

Title: A promising material for bone repair: PMMA Bone Cement Modified by Dopamine-coated Strontium-doped Calcium Polyphosphate Particles
Manuscript ID: RSOS-191028

The editor assigned to your manuscript has now received comments from reviewers. We would

like you to revise your paper in accordance with the referee and Subject Editor suggestions which can be found below (not including confidential reports to the Editor). Please note this decision does not guarantee eventual acceptance.

Please submit your revised paper before 14-Aug-2019. Please note that the revision deadline will expire at 00.00am on this date. If we do not hear from you within this time then it will be assumed that the paper has been withdrawn. In exceptional circumstances, extensions may be possible if agreed with the Editorial Office in advance. We do not allow multiple rounds of revision so we urge you to make every effort to fully address all of the comments at this stage. If deemed necessary by the Editors, your manuscript will be sent back to one or more of the original reviewers for assessment. If the original reviewers are not available we may invite new reviewers.

RSC Associate Editor:
Comments to the Author:
(There are no comments.)

RSC Subject Editor:
Comments to the Author:
(There are no comments.)

Reviewers' Comments to Author:

Reviewer: 1

Comments to the Author(s)

The manuscript describes coating of DOPA as an additional layer on SCCP/PMMA bone cements to improve biocompatibility and stability issues of bone cements that can be used in bone repair. The production method and characterization data is satisfactory and the manuscript can be published in RSOS. However, there minor points to consider about the figures.

Figure 4, 5, 8 is too small to be properly seen. Statistically significant data sets are difficult to be observed from the figures, too.

Figure 7 is it a SEM or confocal microscopy image? Scale bars are impossible to be seen.

Reviewer: 2

Comments to the Author(s)

The novelty of this manuscript is sort of low. Authors lacked of tests in characterizing the synthesized materials and showing enough biological effects, leaving a superficial analysis far from completion. Animal models should be examined and included in this study as in vitro cell study could not convince the suitability of this materials for bone repair. For this reason, I do not see many novelty in this manuscript.

Reviewer: 3

Comments to the Author(s)

Review of "A promising material for bone repair: PMMA Bone Cement Modified by Dopamine-coated Strontium-doped Calcium Polyphosphate Particles" for Royal Society Open Science:

This manuscript reported a new type of biofunctional composite composing DSCPP PAs and PMMA matrix, and evaluated its physicochemical properties and bio-activity. It requires some revisions before acceptance. Below are some of my comments.

1. In the summary, "...the reaction temperature was found to reduce with an increase..." should be "...the reaction temperature was found to be reduced with an increase...". Please also check the grammar in the context.
2. In the introduction, the authors stated that 8% SCPP was used, while in the material section, 5%, 10%, 15% were tested. These information is inconsistent.
3. Image resolution needs improving. For example, it is difficult to read the labels in Fig. 4.
4. The authors inferred that the D/SCPP/PMMA composite bone cements had better mechanical properties than the SCPP/PMMA composite bone cements. However, in Fig 5, their flexural strength/modulus is similar.
5. In the results, 15% doping was demonstrated to be the optimal for the biological activity. However, in the summary section, the authors stated that "the biological and physicochemical properties of this 8% SCPP particles/PMMA composite bone cement were systematically researched".

Author's Response to Decision Letter for (RSOS-191028.R0)

See Appendix A.

RSOS-191028.R1 (Revision)

Review form: Reviewer 1 (Gamze Kuku)

Is the manuscript scientifically sound in its present form?

Yes

Are the interpretations and conclusions justified by the results?

Yes

Is the language acceptable?

Yes

Do you have any ethical concerns with this paper?

No

Have you any concerns about statistical analyses in this paper?

No

Recommendation?

Accept as is

Comments to the Author(s)

After author corrections, the manuscript is appropriate to be published in Royal Society Open Science.

Decision letter (RSOS-191028.R1)

02-Sep-2019

Dear Professor yu:

Title: A promising material for bone repair: PMMA Bone Cement Modified by Dopamine-coated Strontium-doped Calcium Polyphosphate Particles

Manuscript ID: RSOS-191028.R1

It is a pleasure to accept your manuscript in its current form for publication in Royal Society Open Science. The chemistry content of Royal Society Open Science is published in collaboration with the Royal Society of Chemistry.

RSC Associate Editor:
Comments to the Author:
The manuscript can now be accepted as is.

RSC Subject Editor:
Comments to the Author:
(There are no comments.)

Reviewer(s)' Comments to Author:
Reviewer: 1

Comments to the Author(s)
After author corrections, the manuscript is appropriate to be published in Royal Society Open Science.

Appendix A

Dear Editor,

We would like to thank Royal Society Open Science for giving us the opportunity to revise our manuscript “A promising material for bone repair: PMMA Bone Cement Modified by Dopamine-coated Strontium-doped Calcium Polyphosphate Particles”, ID: RSOS-191028.

We thank the reviewers for their careful read and thoughtful comments on previous draft. We have carefully taken their comments into consideration in preparing our revision. The following are the responses to the comments.

Sincerely yours,

Xixun Yu

Reviewer: 1

Major comments

1. Figure 4, 5, 8 is too small, statistically significant data sets are difficult to be observed.
2. Figure 7 is a SEM or confocal microscopy image? Scale bars are impossible to be seen.

Response

Thank you very much for your comments. Your comments are critical to this manuscript. Below are our responses to your valuable comments:

1. We are very sorry for this mistake. The size and resolution of all images have been revised, including Figure 4, 5, 8.
2. Figure.7 is the confocal microscopy image, not the SEM image. We wrote a wrong legend for this figure and it has been revised in revised manuscript. Each scale bar is 200um. The image size must be reduced due to limitations on space, therefore, the scale bar is also reduced and not easy to be seen.

Reviewer: 2

Major comments

The novelty of this manuscript is sort of low. Authors lacked of tests in characterizing the synthesized materials and showing enough biological effects, leaving a superficial analysis far from completion. Animal models should be examined and included in this study as in vitro cell study could not convince the suitability of this materials for bone repair. For this reason, I do not see many novelty in this manuscript.

Response

Thank you for your suggestion, all your suggestions are very important, they have important guiding significance for this manuscript and our future scientific research. Below are our responses to your valuable comments:

1. Although some ceramic particles (including hydroxyapatite, tricalcium phosphate, SiO₂, etc.) are added into PMMA to improve the performance of PMMA, the biological effects of these particles are considerably lower than SCPP particles used in this study. Our previous studies showed that SCPP can greatly promote the growth of osteoblasts, and it also had a good stimulatory effect on the secretion of angiogenic growth factors (including VEGF and bFGF) by

osteoblasts. All these characteristics are very important for the practical application of bone cement. No literature report about adding SCPP particles into PMMA to prepare the ceramic particles/PMMA composite bone cements were reported as far as we known. Therefore, our researches have some novelty. Furthermore, miraculous biomolecule DOPA was introduced to modify the surface of ceramic particles to improve the interfacial compatibility with PMMA in this study. This method has not been reported in preparing the ceramic particles/PMMA composite bone cements. It also shows some novelty.

2. We carried out a series of tests in characterizing the synthesized materials in this paper. However, most of the characterization results are presented in supplementary materials due to limitations on space, including:

(1) We used EDS and XPS to confirm whether dopamine was successfully coated on SCPP surface, or whether the SCPP and D/SCPP particles were successfully doped into the PMMA matrix. (Supplementary Figure 1-2).

(2) We used SEM and automatic specific surface & pore analyzer to observe the cross-section morphology of the synthesized materials and detect the porosity and pore size distribution of the synthesized materials. (Supplementary Table 2, Supplementary Figure 3, Figure.2).

(3) We used Micro-CT to confirm whether D/SCPP and SCPP were evenly distributed in PMMA (Figure 3).

(4) The in vitro mineralization performance of the synthesized materials was also tested in this paper. (Supplementary Figure 4-5, Supplementary Table 3).

(5) The physicochemical properties (Figure 4) and mechanical properties (Figure 5) of the synthesized materials were also been characterized.

These test results indicated that dopamine and strontium elements were successfully incorporated into PMMA, and the physicochemical properties and mechanical properties of the synthesized materials were improved.

3. As regard to the biological effects of the synthesized materials, this paper mainly verified the cytocompatibility of these materials, including:

(1) The cell proliferation rate on materials (Figure 6).

(2) Observing the adhesion and growth of cells on the materials by confocal microscopy and SEM (Figure 7, Supplementary Figure 6)

(3) To further research the effect of the synthesized materials on osteogenesis process, the human bone morphogenetic protein 2 (BMP-2), vascular endothelial growth factor (VEGF) and alkaline phosphatase (ALP) which secreted from cells co-cultured with synthesized materials were also determined using ELISA.

The results demonstrated that the synthesized materials were beneficial to the cell adhesion and proliferation, and the expression of related cytokines which were in favour of the process of osteogenesis, thus we can infer the positive effects of the materials on bone repair from the cellular level.

4. About animal experiment

I am very grateful to reviewer for your suggestion because animal experiments are indeed an important experiment to convince the suitability of the materials for bone repair. Our research group have demonstrated that 8% SCPP particles could promote bone repair and prevent the aseptic loosening of artificial joints in vivo in the early time. The animal experiments of these synthesized materials (SCPP/PMMA and D/SCPP/PMMA) have been designed, and it is expected

to start in November of this year in our following studies. Some results of the animal experiment of 8% SCPP about bone repair-promotion and aseptic loosening-prevention of artificial joints are presented as follows:

Figure.1 Material implantation procedure

- (S1): Expose the knee joint and the intercondylar fossa, with the intercondylar fossa as the surgical entrance;
- (S2): Use a 2.8mm diameter bone drill to drill a hole of 2.8*18mm;
- (S3): Use gauze to stop bleeding until there is no more blood seepage;
- (S4): Implanting the titanium rod prosthesis of the corresponding material of the surface coating into the bone hole;
- (S5): Use bone wax to close the bone hole;
- (S6): Surgical area tissue sutured layer by layer;

Figure.2 X-rays of knee-joint (G1: 8%SCPP+UHMWPE+Titanium rod; G2: UHMWPE+Titanium rod; G3: Titanium rod) At 20w, we can clearly see by X-ray that the osteolysis area around the titanium rods in the G1 and G2 groups is larger than that in the G3 group, and the G2 group is slightly larger than the G1 group;

Figure.3 Femoral scans by micro-CT (G1: 8%SCPP+UHMWPE+Titanium rod; G2: UHMWPE+Titanium rod; G3: Titanium rod) As shown in the 2D image, the new osteoid in the surface of the bone-titanium rod prosthesis in G1 group was more than the G2 and G3 groups; the 3D reconstruction showed that the bone morphology and bone density of around the bone-titanium prosthesis in the G1 group were better than G2 and G3 group.

Reviewer: 3

Major comments

1. some grammar errors like “was found to reduce” .
2. The meaning of 8%, 5%, 10%, 15%.
3. Image resolution.
4. The flexural strength/modulus of D/SCPP/PMMA and SCPP/PMMA is similar.
5. “15% doping” and “8% SCPP”

Response

We appreciate very much the reviewer’s comments. The comments will be beneficial to the improvement of the manuscript and our future research. Below is our response to your valuable comments:

1. We apologize for this error, “was found to reduce” has been changed to “was found to be reduced” , and we have corrected it in revised manuscript.
2. 8% (SCPP) means the mass percent of Sr element doped in CPP is 8%. Previous studies in our researches showed that the biological effect of 8% SCPP was the best, so the 8% SCPP particles were employed as the bioactive ceramic fillers to modify PMMA bone cements. 5%, 10%, 15% are refer to the weight fractions of 8% SCPP particles admixed to PMMA. We set these three concentration gradients to find out which weight fraction of 8% SCPP particles is the optimal proportion in preparing the ceramic particles/PMMA composite bone cements.
3. The size and resolution of all images in the manuscript have been revised.
4. The flexural strength/modulus of D/SCPP/PMMA is only a little higher than SCPP/PMMA, and the difference is not significant. In terms of mechanical testing, this is an imperfect experimental result. However, the cytocompatibility of D/SCPP/PMMA is better than that of SCPP/PMMA, which is also the reason why dopamine is introduced to improve SCPP/PMMA cytocompatibility in this experiment.
5. 8% (SCPP) means the mass percent of Sr element doped in CPP is 8%. 15% is refer to the weight fraction of 8% SCPP particles admixed to PMMA. The results of cytocompatibility test, including the adhesion and proliferation of cells on the surface of the materials and the expression of ALP, BMP-2 and VEGF from cells co-cultured with synthesized materials, showed that the best

weight fractions of 8% D/SCPP particles admixed to PMMA was 15%.

In addition, due to a spelling mistake, the given name of second co-author is wrong. The correct should be “can”, not “chan” . The second co-author’s given name has been corrected in revised manuscript.